# Managerial Practices and (Post) Pandemic Consumption of Private Labels: Online and Offline Retail Perspective in a Portuguese Context

**Juliana Pires Pinto [1], Cláudia Miranda Veloso [2],\*[ID], Bruno Barbosa Sousa [3],\*[ID], Marco Valeri [4][ID], Cicero Eduardo Walter [5][ID] and Eunice Lopes [6][ID]**

1   ESTGA, University of Aveiro, 3810-193 Aveiro, Portugal
2   GOVCOPP, ESTGA, University of Aveiro, 3810-193 Aveiro, Portugal
3   CiTUR, Polytechnic Institute of Cávado and Ave (IPCA), 4750-810 Barcelos, Portugal
4   Faculty of Economics, Niccolò Cusano University, 00166 Rome, Italy
5   GOVCOPP, Department of Economics, Management, Industrial Engineering and Tourism, University of Aveiro, 3810-193 Aveiro, Portugal
6   TECHN&ART, Polytechnic Institute of Tomar (IPT), 2300-313 Tomar, Portugal
\*   Correspondence: cmv@ua.pt (C.M.V.); bsousa@ipca.pt (B.B.S.)

**Abstract:** Currently, given the different dynamics of competition, food retailers are increasingly betting on private labels as a strategy of differentiation and retention of competitive advantages. To this extent, this study aims to assess the antecedents of the purchase intention of food retailers' private labels, as well as to understand the managerial practices and (post) pandemic consumption retail perspective in a Portuguese context. The results obtained, through a quantitative analysis by means of multiple linear regressions, on a random sample of customers (*n* = 300) indicate that customer satisfaction and attitude towards his/her own brand are quite favourable, as demonstrated by the existence of a high loyalty to his/her own brand. Additionally, they reveal that the purchase experience, the private-label image, the perceived risk, and the COVID-19 pandemic are prior attitudes towards the private label and its mediation in the purchase intention and recommendation of its products. Finally, loyalty to the private label, store satisfaction and, through these, also the shopping experience were confirmed as determinants of loyalty. These results provide insights to food retailers on aspects to be improved and considered in the design of commercial strategies that promote the intention to purchase private-label products and that win and retain customers and achieve competitive advantages and profitability. Regarding the COVID-19 pandemic, the study reveals that some consumers changed their purchasing patterns, choosing to buy more private-label products at this stage.

**Keywords:** private labels; perceived risk; COVID-19 pandemic; loyalty; retail; consumer sustainability; online shopping; offline shopping

## 1. Introduction

In an increasingly competitive and dynamic market, retail companies are interested in obtaining differential advantages compared to their competition. In order to attract as many customers as possible, they apply private-label strategies to offer them greater added value. To achieve this differentiation, it is essential that retailers understand the most relevant factors to consumers when purchasing private-label products. Private brands play an important role for consumers, as they constitute a testimony to those who use or experience them, being used in a certain context or social environment [1]. Another important point of private labels is the impact on people and the values that embody them, becoming much more than just products/services. Keller and Swaminathan [2] argue that private labels are a source of competitive advantage and a source of financial returns. Aaker, Biel and

Biel [1] add that the value that a brand represents for a company is indescribable due to the difficulties that exist in building and establishing brands, which is a result of the high costs of advertising, distribution, and also the high proliferation of brands in the market. The adoption of new strategies by retailers is linked to the improvement of the purchasing process, adopting a simplified and personalized service capable of satisfying the needs of any type of customer. According to the Private Label Manufacturers Association—PLMA (2021), in Portugal, private labels have shown exponential growth, representing a market share corresponding to 45%, well above the European average (38%). In recent decades, investigations into private labels have been the focus of study by researchers from all over the world e.g., [3–5]. The fact that the market is increasingly competitive and that there is a growing need for more economical offers for various products make this topic very current and under constant study. In this study, we seek to deepen the knowledge about private brands and the surrounding factors capable of influencing consumer behavior and loyalty [6].

However, despite the vast amount of research on this topic, it is not a topic that gathers a consensus on the part of researchers. Given the changes in the world of retail in general and in consumption patterns, it is difficult to keep up with the needs and preferences of consumers, so both researchers and retailers try to bring together as many influencers as possible so that the response is increasingly agile and effective. With this study, it is intended to evaluate a set of antecedents approached by other authors in isolation, aggregating them into the same conceptual model that allows one to carry out a compilation so that the study is the most adequate and complete for organizations. It is also intended that the model be statistically validated through the analysis of multiple linear regressions. The objectives of this study focus on the perception of dimensions that consumers take into account when purchasing private-label products (with special emphasis on sustainability in consumption). After this introduction, there is a presentation of the literature review of the main background, then the conceptual model and the research hypotheses are proposed, followed by the presentation and discussion of the results, ending with the conclusion and contribution of the research.

## 2. Background

### 2.1. Private-Label Image

A brand's image is the sum of all consumer memories arising from their perceptions of a brand. The image of private labels is formed by two dimensions, the affective one and the quality of the brand. The authors of this paper identified that the impacts of the store image on the affective dimension of private labels include quality, convenience and the perception of price versus value. They claim that these are the dimensions that positively influence the consumer's attitude towards private labels [7]. Van Loo et al. [8] concluded that for brands that have a better image, consumers have a positive attitude and greater purchase intention. Modern day store brands (SB) or private labels (PL), now also normally called private brands, are the ones generally owned and marketed by retailers. They have been active on the market for about 70 years. Over this time span, these brands have evolved from generic, cheap, low-quality economy or budget private labels to lower-priced-than-national brands whilst still being acceptable-quality value or standard private labels [9].

**Hypothesis 1a (H1a).** *The image of private labels has a direct and positive impact on the behavioral intention for the private labels.*

**Hypothesis 1b (H1b).** *The image of private labels has a direct and positive impact on attitudes towards the private labels.*

### 2.2. Sustainable Shopping Experience

Value can only be created through its usefulness or emotionality. According to the [10] study, the shopping experience can result in value through the performance of a utilitarian task or a hedonic pleasure. The quality, price, and emotional values of purchase value have

a positive impact on store loyalty. Utility values help consumers increase the purchase acquisition utility and increase the efficiency of the shopping experience through some conditions such as economy, quality, and convenience. Consumers choose products that provide them with a high level of value and benefit [11].

Hedonic value is defined as entertainment and it is associated with purchase motives such as pleasure, enjoyment, aesthetics, and fun [12]. Hedonic values are associated with emotions such as pleasure–entertainment, interest–curiosity, surprise–surprise, among others. Thus, both the utilitarian value and the hedonic value, translated into the shopping experience, are related to customer attitudes towards private labels and they are also related to loyalty to the private labels. Therefore, it also significantly contributes to retailer loyalty and to positively influencing the store image [13]. With the rapid development of online shopping and traditional physical store shopping interweaving to form different shopping situations, customer experience has gradually become the main source of retailers' sustainable competitive advantage through differentiation. Retailers need to continuously improve the customer experience in different shopping situations to maintain long-term sustainable customer satisfaction and achieve sustainability [14].

**Hypothesis 2a (H2a).** *The shopping experience positively affects the image of private labels.*

**Hypothesis 2b (H2b).** *The shopping experience positively affects attitudes towards private labels.*

**Hypothesis 2c (H2c).** *The shopping experience positively affects store satisfaction.*

**Hypothesis 2d (H2d).** *The shopping experience positively affects store loyalty.*

**Hypothesis 2e (H2e).** *The shopping experience negatively affects perceived risk.*

*2.3. Perceived Risk*

When the theme is private labels, the perceived risk is considered a relevant determinant, especially when comparing private label alternatives with suppliers' brands [15]. Consumers see suppliers' brands as safer brands with less variation in quality than private brands. Risk can manifest itself in different ways, such as the fear of a product not having desirable attributes or the uncertainty about the products performance. In general, private brands are marketed at lower prices, which can compromise brand image. The perceived risk tends to be lower when prices are higher, as consumers follow a relationship between price and quality [16].

Financial risk is the possibility of monetary loss caused by poor choice in the purchase process. Perceived risk influences the behavior and purchase intention of their own brands. Brand familiarity reduces risk, that is, the greater the familiarity with a private label, the smaller the difference between this brand and the supplier brand in terms of perceived risk.

**Hypothesis 3a (H3a).** *Perceived risk has a direct and negative impact on the image of the private label.*

**Hypothesis 3b (H3b).** *Perceived risk has a direct and negative impact on behavioral intent to the private label.*

*2.4. Attitude Regarding the Private Brand*

Attitude is based on a set of information regarding the evaluated object and is progressively accumulated by the individual (cognitive component). Attitude is oriented, as it expresses a positive or negative evaluation in relation to the object (affective component). Attitude is dynamic and it is a predisposition to action; as such, it is a predictor of behavior (behavioral component). In this sense, we can conclude that attitudes are predispositions that are reflected in purchase options.

However, there are three main factors that help one to understand the relationship established between consumers and brands according to [17]: (1) Sensitivity to brands, i.e., a consumer is sensitive to a brand when looking for information about it, the information found being a decisive factor in their choice; (2) brand loyalty indicates the degree of preference, more or less exclusively, for a brand in the course of the purchase process; (3) the nature of the purchase, i.e., the purchasing attitude can be, as the case may be, methodical, thoughtful, or impulsive.

The most important factors in a brand are mostly the quality/price ratio, quality, and finally price. They claim that the reasons that lead respondents not to consume their own brand are the lack of quality, distrust, the poor image of the store, and the brand image. The attitude towards private labels is defined as the predisposition to respond favorably or unfavorably to the evaluation of a private-label product. The purchase intention is directly influenced by the attitude of consumers towards the private label, that is, the more favorable, the stronger the purchase intention should be.

**Hypothesis 4 (H4).** *Attitudes towards private labels directly and positively affect the behavioral intention towards the private labels.*

### 2.5. COVID-19

With the pandemic crisis that affected the entire world, the strength and relevance of private labels was even more reinforced. It is true that private label sales grow in times of crisis, and the COVID-19 pandemic has come to authenticate this [18]. Customers started to use the verb "need" rather than "want", proof of this was the rush to some basic necessities at an early stage given the fear of disruptions to these products. People are increasingly choosing to buy their own brands. Thus, consumers are able to guarantee savings, quality, and security at a time of crisis and instability such as the one we are going through.

In general, consumers in a situation of pandemic crisis choose to change their consumption patterns and resort to new solutions that are more practical and economical [19]. It is estimated that the percentage of customers who started looking for more private-label products in a trip to the supermarket increased, however, some studies predict that most of these people will maintain these new consumption habits after the pandemic crisis.

**Hypothesis 5a (H5a).** *The COVID-19 pandemic has a direct and positive impact on the shopping experience.*

**Hypothesis 5b (H5b).** *The COVID-19 pandemic has a direct and positive impact on attitudes towards private labels.*

**Hypothesis 5c (H5c).** *The COVID-19 pandemic has a negative impact on perceived risk.*

### 2.6. Behavioral Intent Regarding Private Brands and Behavioral Intentions

Service quality significantly influences behavioral purchase intentions [20]. Through previous researchers, an association between the quality of the service and particular dimensions of the behavioral intention has been verified. Loyal customers are the main assets of companies, it is supposed that positive perceptions of service quality increase the possibility of customers dedicating themselves to supporting the company and developing and reinforcing loyalty behavior [21].

The marketing literature is consensual in the statement that a good and positive brand image has a positive effect on the customer's behavioral intentions towards private labels, enabling the retailer to increase the number of loyal customers to the store and generate a more favorable and positive recommendation. Customers have a positive attitude and a greater intention to buy products from brands that have a good reputation and a better image.

**Hypothesis 6a (H6a).** *The behavioral intention for private labels has a direct and positive impact on store satisfaction.*

**Hypothesis 6b (H6b).** *The behavioral intent of private labels has a direct and positive impact on store loyalty.*

Customer satisfaction regarding the store must be one of the main concerns of companies, since satisfied customers are loyal customers, becoming an essential step for the success of any organization [22]. It appears that customer satisfaction depends on the quality of service provided to the customer and is one of the value-added instruments. Companies exist because they have a customer to serve. The key to customer satisfaction lies in the challenge of identifying and anticipating their needs and, especially, in the ability to satisfy them. For customers to increase their loyalty, the perceived value of the good or service presented must be at the level of reality, making it an integral part of the organization's corporate objectives. To be successful, especially in the retail industry, managers must focus on customer retention, implementing effective strategies for customer satisfaction and loyalty, since the cost of attracting new customers is higher than the cost of keeping current ones [23].

**Hypothesis 7 (H7).** *Store satisfaction directly and positively influences retailer loyalty.*

According to [23], store loyalty is defined as the tendency to repeat purchases of similar products whether or not in the same store, with loyal consumers spending a large part of their expenses in the store where they are loyal. There is a positive correlation between the use of private labels and store loyalty, that is, the fact that consumers spend much of their time and money on a single chain increases exposure, familiarity, and willingness to buy private labels from that store. Retailer employees are also key elements, as they are able to influence customer loyalty through excellent service delivery.

**3. Methodology**

According to the literature review and the hypotheses already postulated, the aim of this study is to propose a conceptual model (Figure 1) that consists of eight main constructs: purchase experience, private brand image, perceived risk, COVID-19 pandemic, attitudes towards private label, behavioral intent to the private label, store satisfaction, and store loyalty.

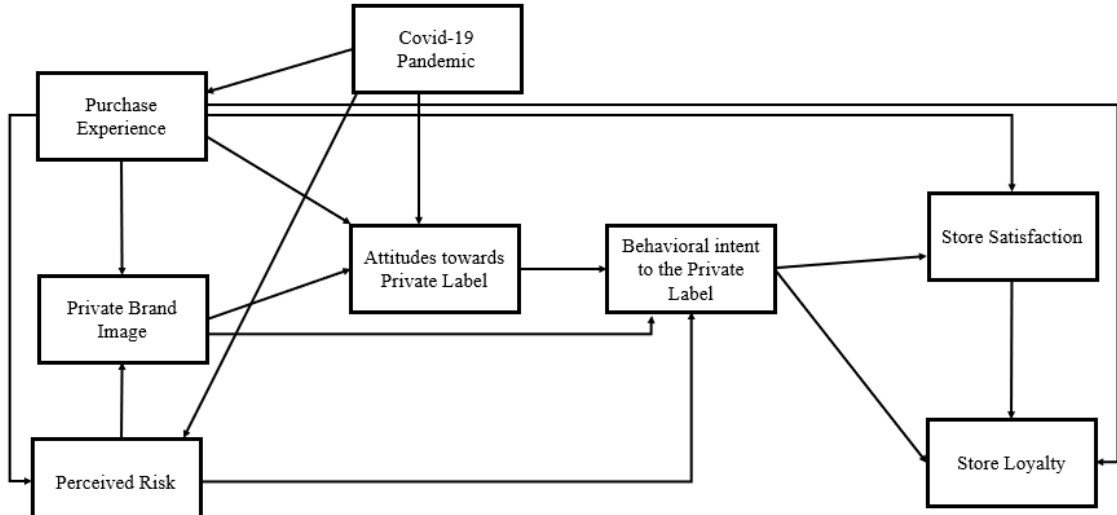

**Figure 1.** Conceptual Model.

In this study, a quantitative method approach was followed, which consisted of developing questionnaires, sampling, data collection, and statistical data analysis. The

questionnaire was divided into ten sections—eight dimensions, purchase process, and personal data, and it was subjected to a semantic analysis of ten members with experience in retail and customer support to ensure the validity of the questionnaire content. The questionnaire contained 49 items in total, in which the purchase experience, private brand image, perceived risk, COVID-19 pandemic, attitudes towards private brand, behavioral intention towards private label, store satisfaction and store loyalty were measured (Table 1). The questionnaire questions were measured on a 7-point Likert scale, where 1 represented "Strongly Disagree" and 7 "Strongly Agree".

**Table 1.** Survey questions.

| SURVEY QUESTIONS | |
|---|---|
| **Store Satisfaction** | I'm satisfied with this hypermarket. |
| | Considering what I think is an ideal hypermarket, considering that an Auchan is ideal for me. |
| **Purchase Experience** | I was able to find/buy all the products I needed. |
| | I was happy because I don't need to go to other store(s) to fulfill my purchases. |
| | While shopping, I had a feeling of new experience. |
| | This shopping trip was a good time. |
| | In addition to the products purchased, I really enjoyed the shopping experience. |
| | I continued to buy, not out of necessity, but out of desire and because I felt good at the store. |
| | Compared to other activities I had planned; the shopping trip was really enjoyable. |
| **Perceived Risk** | If I buy Auchan brand products, I will spend my money well. |
| | If I buy private brand products, I believe the financial investment was well made. |
| | When considering choosing private brand products, I am reassured that the product has the level of performance and quality I expect. |
| **Private Brand Image** | I'm satisfied with most of the private brand products I buy at Auchan. |
| | I like Auchan's private brand. |
| | The private brand products I buy at this store have good quality. |
| | I think that Auchan cares about the quality of its private brand products. |
| | The private brand products I buy from this store have a good level of performance. |
| **Attitudes toward Private Label** | I look for private brands when I go shopping. |
| | When I buy private brands, I feel I'm getting a good deal. |
| | For most products categories, the best purchase option is usually Auchan private brand. |
| | Considering the cost-benefit ratio, I prefer private brands to other brands (manufacturers' brands). |
| | My shopping cart contains private brands of several products. |
| **COVID-19 Pandemic** | The current al economic situation has led you to buy a wider variety of private brand products rather than supplier brands. |
| | I have changed my buying patterns due to the pandemic and economic situation in the country. |
| **Behavioral Intent to the Private Label** | I intend to buy private brand products from this store frequently. |
| | There is a strong possibility of buying private brand products. |
| | I would recommend the purchase of Auchan private brand products to my family and friends. |
| **Store Loyalty** | Auchan is my first choice when I think about shopping. |
| | I intend to continue shopping at this hypermarket. |
| | I would recommend Auchan Aveiro to my family and friends. |

This study was applied to customers of the Auchan store in Aveiro and it was disclosed in person, through flyers and online, via a link, and via QR code. The sample collected was of 300 valid respondents. The IBM SPSS Version 24 software program was used to carry out the statistical treatment. To systematize and highlight the information provided by the data, descriptive statistical techniques were used. Variables were worked on and recoded. To characterize the results, both relative and absolute frequency tables and statistical measures

were used: minimum, maximum, mean, standard deviation, and variance. The internal consistency of the scale's dimensions and sub-dimensions was also measured, and the Cronbach's alpha coefficient was used, which ranged between 0.850 and 0.940, suggesting that the construction could be used with confidence. In order to measure the intensity of the linear correlation between the variables, Spearman's correlation coefficient was used. Multiple linear regression was used to estimate models that could identify the determinants of the behavior of the variables under analysis. Throughout the analysis and for decision making, a significance level of 5% was assumed.

## 4. Data Analysis and Discussion

The sample collected comprised a total of 300 participants, 26% (78) of which were male and the rest female. The maximum number of responses was obtained in the range of 36 to 45 years of age with 40% (120) of the total responses. The majority of respondents were married with 55.7% (167). It was observed that most of these had higher education qualifications, since 52% (156) revealed in their answer that they were graduates. It appeared that the most represented household was made up of 4 elements, represented by 34% (102). As for professional occupation, the majority of 66% (198) were employees. In relation to the household's net monthly income, it was observed that 24.3% (73) earned between 750 and 1250 euros monthly. Finally, most respondents resided in the municipality of Aveiro. In the purchase process, it appeared that the majority of Auchan's customers, 68% (204), had been shopping at this hypermarket for more than five years. Regarding the frequency of purchases, it was found that 26.3% (79) of Auchan's customers made purchases five or more times a month. As for the monthly average of spending on purchases, it was found that 19.7% (59) of respondents spent between 51 to 100 euros. Regarding the type of purchase, the vast majority, 81% (243) of respondents said that it was in this store that they most frequently shopped for consumer products such as groceries, fruits, toiletries, beverages, frozen goods, and others. As mentioned, Auchan has its own brands in different markets, and it was found that more than half of the respondents knew the following private brands: Auchan, Cosmia, Qilive, and Actuel. As for anchor products (products that stand out in a brand since their main characteristic is to make consumers visit the store), we observed that, due to the different responses, this insignia did not contain anchor products. The main reason for most respondents, 93.3% (280), choosing to purchase private-label products was price, followed by the quality and exclusivity of private-label products. Regarding the markets in which respondents chose to purchase more private-label products, it was clear that the most-preferred markets were the salted grocery store with 71.3% (214) of the responses, followed by the sweet grocery store with 62.3% (187). Due to the pandemic situation experienced worldwide, 67% (201) of respondents said they have chosen to use the various channels made available by Auchan. Regarding the channels most used by customers, we found that traditional in-store purchases were still the most-used method in the purchase process, where 69% (207) of respondents revealed their preference, however, 54.3% (163) of consumers opted for online shopping with delivery.

In the first regression model presented, the influence of the COVID-19 pandemic on the variation in the shopping experience was tested. The model obtained at a significance level of 5% was statistically significant, that is, the variation in the shopping experience was significantly explained by the estimated model. By applying the t test, it was concluded that the COVID-19 pandemic ($\beta = 0.285$; $p < 0.001$) significantly determined the behaviour of the shopping experience. The coefficient of determination revealed that the model presented explains, on average, about 7.8% of the variation in the shopping experience. According to this model, Hypothesis H5a is validated.

The second estimated model relates the image of the private label with the regressors, i.e., purchase experience and perceived risk at a statistically significant 5% significance level. The determinants of the model that significantly influenced the variation in the brand image were by the t test: shopping experience with coefficient $\beta = 0.026$ and $p < 0.001$ and the perceived risk with coefficient $\beta = -0.901$ and $p < 0.001$. The estimated model explains,

on average, around 84.6% of the variation in the image of the private label. According to this model, Hypotheses H2a and H3a are accepted.

The third estimated model relates the perceived risk with the following regressors: purchase experience and COVID-19 at a statistically significant 5% significance level. By performing the t test, it was verified that the shopping experience ($\beta$ = −0.723 and $p < 0.001$) and COVID-19 ($\beta$ = −0.036 and $p$ = 0.001) negatively and significantly influenced the variation in perceived risk. The estimated model explains, on average, around 53.5% of the variation in customer satisfaction. In accordance with this estimated model, Hypotheses H2e and H5b are corroborated.

The fourth estimated model relates the attitudes towards private labels with the following regressors: private brand image, shopping experience, and COVID-19 at a statistically significant 5% significance level. The determinants of the model that significantly influenced the variation in attitudes towards the private brand were shopping experience with coefficient $\beta$ = 0.187 and $p$ = 0.001; own-brand image with $\beta$ = 0.5445 and $p < 0.001$; and COVID-19 with $\beta$ = 0.135 and $p$ = 0.001 coefficient. The estimated model explains, on average, about 53.4% of the variation in attitudes towards private labels. According to the model, the following hypotheses are authenticated: H1a, H2b, and H5c.

The fifth estimated model relates the behavioural intention related to the private brand with the following regressors: attitudes towards the private brand, perceived risk, and image of the private brand at a statistically significant 5% significance level. By applying the t test, it was concluded that attitudes towards private labels ($\beta$ = 0.514; $p < 0.001$) and the image of the private labels ($\beta$ = 0.379; $p < 0.001$) significantly determined the behaviour of relative behavioural intention to the own brand. Perceived risk did not have a significant influence on the variation in behavioural intention related to the private labels. The estimated model explains, on average, about 72.4% of the variation in behavioural intention related to the private labels. In harmony with the estimated model, Hypotheses H1b and H4 can be validated.

The sixth estimated model relates store satisfaction with the following regressors: purchase experience and behavioural intention related to the private labels at a statistically significant 5% significance level. The determinants of the model that significantly influenced the variation in store satisfaction, verified through the t test, were the shopping experience with a coefficient $\beta$ = 0.617 and $p < 0.001$ and the behavioural intention related to the private labels with a coefficient $\beta$ = 0.192 and $p < 0.001$. The estimated model explains, on average, around 54.9% of the variation in store satisfaction. According to the estimated model, Hypotheses H2c and H7a are accepted.

Finally, the seventh estimated model relates store loyalty with store satisfaction, with the behavioural intention related to the private labels and with the shopping experience at a statistically significant 5% significance level. By applying the t test, it was concluded that store satisfaction ($\beta$ = 0.549; $p < 0.001$), behavioural intention related to the private labels ($\beta$ = 0.182; $p < 0.001$), and the shopping experience ($\beta$ = 0.126; $p$ = 0.028) significantly determined the behaviour of store loyalty. The estimated model explains, on average, about 57.9% of the variation in store loyalty. In accordance with the estimated model, the Hypotheses H2d, H6, and H7b are validated.

Table 2 presents a summary of the analysis of the hypotheses under study. These hypotheses were a consequence of the proposed objectives. With the application of the multiple linear regression model, the results obtained allowed us to obtain a model that provided the validation of all Hypotheses except for the H3b hypotheses.

In accordance with the analysis performed using multiple linear regressions, it appears that all of the research hypotheses postulated in this study were validated, with a significance of at least $p < 0.05$, except for the third hypothesis, which was partially validated. In the evaluation of the third hypothesis, the perceived risk, through the analysis of multiple linear regressions, indicated a negative effect on the client's behavioural intention in relation to the private labels. Not having proved to be significant, this result is in line with what has been shown in previous studies. As a result, it is believed that these results can

be justified by the fact that the perceived risk, both financial and performance, of Auchan private-label products is relatively low, which does not negatively affect the sustainable behavioural intention of buyers of private-label products.

**Table 2.** Data Analysis Summary.

| Hypothesis | Research Hypothesis | β | *r* | Supported Hypothesis? |
|---|---|---|---|---|
| H1a | Private Brand Image → Attitudes towards Private Label | 0.545 | <0.001 | **Yes** |
| H1b | Private Brand Image → Behavioral Intent to the Private Label | 0.379 | <0.001 | **Yes** |
| H2a | Purchase Experience → Private Brand Image | 0.026 | <0.001 | **Yes** |
| H2b | Purchase Experience → Attitudes towards Private Label | 0.187 | 0.001 | **Yes** |
| H2c | Purchase Experience → Store Satisfaction | 0.617 | <0.001 | **Yes** |
| H2d | Purchase Experience → Store Loyalty | 0.126 | 0.028 | **Yes** |
| H2e | Purchase Experience → Perceived Risk | −0.723 | <0.001 | **Yes** |
| H3a | Perceived Risk → Private Brand Image | −0.901 | <0.001 | **Yes** |
| H3b | Perceived Risk → Behavioral Intent to the Private Label | −0.030 | 0.712 | **No** |
| H4 | Attitudes towards Private Label → Behavioral Intent to the Private Label | 0.514 | <0.001 | **Yes** |
| H5a | COVID-19 Pandemic → Purchase Experience | 0.285 | <0.001 | **Yes** |
| H5b | COVID-19 Pandemic → Perceived Risk | −0.036 | <0.001 | **Yes** |
| H5c | COVID-19 Pandemic → Attitudes towards Private Label | 0.135 | 0.001 | **Yes** |
| H6a | Behavioral Intent to the Private Label → Store Satisfaction | 0.192 | <0.001 | **Yes** |
| H6b | Behavioral Intent to the Private Label → Store Loyalty | 0.182 | <0.001 | **Yes** |
| H7 | Store Satisfaction → Store Loyalty | 0.549 | <0.001 | **Yes** |

This study demonstrated a direct and negatively significant effect between perceived risk and private-label image. To improve the quality of private-label products, as well as to reduce the variation in the quality of products belonging to the same categories and, fundamentally, to establish a good value for money, translated into greater benefit for the customer, these are measures that reduce the perceived risk. Additionally, a direct and negative influence of the perceived risk on the customer's behavioural intention regarding the purchase or recommendation of private-label products was also identified. The perceived risk is composed of a financial component associated with the fear of financial loss in the purchase of private-label products and another component related to the performance of these products, both components having a negative impact on the image of private labels and on the purchase intention and recommendation of branded products, although in this study this effect on purchase intention was not significant. The shopping experience construct, composed of the functional value and hedonic value subdimensions, cooperates to reduce the perceived risk and, conversely, encourages a positive attitude in the customer's attitude towards private-label products. Thus, whenever Auchan implemented actions to improve the usefulness and emotions felt, it reinforced the shopping experience, that is, the feeling of accomplishment of the task and pleasure, and of the fun of buying own-brand products and contributed to the reduction of perceived risk to its own brand. The shopping experience proved to be a positive predictor of customer attitudes towards private-label products and customer satisfaction and loyalty to the retailer, as well as a negative antecedent of perceived risk. Additionally, the study demonstrated that the COVID-19 pandemic also exerted a negative and significant effect on the perceived risk, and to this extent this pandemic has contributed to reducing the perceived risk of customers in the face of own-brand products.

The pandemic context allowed food retail brands to invest in digital channels and technological resources. This topic is strongly related to sustainability (i.e., the sustainable purchasing process and sustainable consumption in rationalizing material purchases). Consumers seem to be strongly interested in digital solutions that favour sustainability and balance in the purchase process (i.e., greater peace of mind in the decision, greater reflection, less pressure, and easier comparison through online solutions). The pandemic context is assumed, therefore, as a determinant in the final consumer's decision process,

favouring socially responsible (and less impulsive) behaviours and attitudes. The results of this study confirmed a direct and positively significant impact between the image of private labels and the attitude, which indicates that the image of private labels has an effect on the formation of attitudes towards the sustainable private label. Additionally, it was also found that the COVID-19 pandemic contributed positively and significantly to the attitude towards private labels. Finally, the results obtained indicate that the purchase intention and recommendation of private-label products had direct and positive effects on customer satisfaction and loyalty to the store. These findings are in line with the conclusions reached in previous studies (i.e., customer's behavioural intentions regarding private labels promote customer satisfaction and loyalty to the store). To this extent, this study confirms that the inclusion of private labels in store assortments contributes positively to fostering customer loyalty to the retailer's store, and in this case, to the company.

## 5. Conclusions, Limitations and Next Steps

The focus of previous studies on customer attitudes has been based on the fact that a consumer with a more favorable attitude towards private-label products not only has a great intention to purchase but also to buy in larger quantities. Food retail companies must adopt sustainable strategies that lead their consumers to successively purchase products from their brands and must focus on creating a favorable attitude from consumers and creating strong incentives for greater intention to purchase the retailer's own-brand products [24–27]. To that extent, and in order to implement a commercial strategy to create competitive advantage, they must know the current antecedents of their own brand attitude and loyalty, as well as the impacts and challenges that the COVID-19 pandemic unleashed on consumer behavior relative to their own brand [28–31]. In this context, this manuscript aimed to evaluate the antecedents of purchase intention for their own brand, as well as to understand the impact of the COVID-19 pandemic on consumer habits and to ascertain the degree of satisfaction and loyalty to their own brand and to the studied hypermarket in Portugal.

The results obtained, through a quantitative analysis, using multiple linear regressions and the structural equation model of a random sample (*n* = 300) of customers, suggest that customer satisfaction and attitude towards Auchan's own brand were favorable as well as demonstrate the existence of a great loyalty to the own-brand and to Auchan Aveiro. Additionally, they reveal that the shopping experience, the image of private labels, and the COVID-19 pandemic were antecedents of attitudes towards private-label products. At the same time, the image of private labels, and the perceived risks and attitudes towards private labels are direct predictors of the purchase intention and recommendation of their products, while purchase experience and the COVID-19 pandemic were indirect antecedents mediated by the attitude towards private labels. Finally, brand loyalty and store satisfaction were direct determinants of customer loyalty to Auchan and were also mediators of indirect predictors of the shopping experience in customer loyalty to sustainable brands. According to [32], a clear understanding of consumer motivation to select a brand is essential for developing a brand portfolio optimization model. In the literature, one of the most important factors for selecting a brand is its perceived hedonic value [33–37] and strategic management [38,39]. Perceived hedonic value refers to the enjoyment and pleasure gained during shopping.

Regarding the purchase process, the majority of the sample revealed that the brand customers were customers for more than 5 years and it appeared that most of them carried out mass consumption purchases, which confirms that they were loyal customers to the store. As for company own brands, we found that there were private brands that needed to be further developed and presented to their customers, namely: InExtenso, OneTwoFun, Cup's, Gardenstar, and Airport. Brands must then adopt strategies so that lesser-known brands are presented to their customers, for example, offering a product of a lesser-known private label in purchases with values above a certain value. Through the questionnaire, it was found that the organization did not have flagship products, since in the respondents' answers there was no consensus on a specific product that stood out and brought consumers

to the store. The main reason why most respondents chose to purchase private-label products was the price, however, it was found that they also took into account the quality and exclusivity of private-label products. It is also known that with the pandemic situation currently being experienced around the world, the various channels offered by brands gained more support, with traditional purchases and delivery being the two preferred channels of company consumers.

Regarding the perceived risk, which involves financial risk and performance risk, it is considered that a company should create shopping experiences, both functional and hedonic, with pleasant private-brand products, along with investments in the attributes that customers have said they want and expect to find in these products similar to the products of intrinsic and extrinsic suppliers, such as the packaging and the price, in short, that they fulfill the promised performance, and the establishment of an adequate and fair quality–price ratio that contributes to a reduction in risk. Regarding the intention to buy and recommend Auchan private-label products and loyalty to the store, it is essential to continue the inclusion of private-label products in the store's assortment, focusing on the quality of private-label products, similar to national brands at competitive prices, as it will encourage the purchase and recommendation of the private-label products, whilst also promoting the attractiveness of the store, and preventing the negative contamination of the corporate brand image with a weak own-brand image. The company guarantees the central objective of the introduction of its own brands, which is to promote customer loyalty to the stores of the retail brand. Regarding the COVID-19 pandemic, some consumers confirmed that they have changed their purchasing patterns and even opted to buy more private-label products at this stage, with sweet and salty groceries, fresh, and frozen products being the most prominent markets.

These results provide data to organizations and retailers on aspects to improve and consider when designing commercial strategies that promote the intention to purchase private-label products and are able to attract and retain customers, leveraging the competitive advantage and profitability of the company and the sustainability of the retail insignia. This study offers considerable contributions to science, branding, and to the food and specialized retail sector, as it offers a robust and innovative conceptual model for predicting private brand loyalty, which can be used as a tool to assess the behavioral attitudes of the customer in relation to private labels in future studies to investigate the antecedents and effects of the experience of purchasing private-label products, and simultaneously, it constitutes a guiding instrument for the implementation of a strategy to create a sustained competitive advantage in the variables influencing loyalty to a brand.

After presenting the conclusions drawn from this study, it is worth mentioning the main limitations of the research. The data collection and the method to reach respondents was a limitation, since the dissemination of the study, having been carried out through a link and QR code, tended to represent a younger sample, so this may be considered restricted although its size is sufficient for the analysis and validation of the study. Furthermore, the conclusions resulting from this study cannot be generalized to all Auchan stores in Portugal, as they only reflect the Aveiro store. The fact that the study was carried out in a pandemic situation may also have influenced the results obtained. In the regressions performed, the normality assumption was not verified, however, it is known according to [6] that if the model perturbations do not have a normal distribution, the results obtained regarding the least squares estimators are not affected at all. In order to proceed with the investigation of this topic, it is advisable to deal with the limitations presented. The study carried out and the implemented practices could be expanded to the other stores in the group in order to maximize the benefits and value provided to its customers. Increasing the sample size will also help in the investigation process and future studies should also carry out structural equation modeling, performing path analysis (SEM-PLS). Another future study that would also be very important would be its repetition in a post-COVID-19 pandemic situation. As for the dimensions used, there are several factors that could be added to future studies, such as the recommendation/opinion by third parties, the current economic situation,

private-label packaging, among others. For future research, it is suggested to carry out a comparative study with other sustainable brands operating in Portugal or for a more advanced study, and to extend the study to other countries where the sustainable brand operates (e.g., consumption of private labels).

**Author Contributions:** J.P.P.—Investigation, Conceptualization and Methodology, C.M.V.—Conceptualization, Methodology and Supervision, B.B.S.—Conceptualization and Writing—original draft, M.V.—Writing–original draft and Resources, C.E.W.—Conceptualization, E.L.—Writing-review & editing. All authors have read and agreed to the published version of the manuscript.

**Funding:** This research received no external funding.

**Conflicts of Interest:** The authors declare no conflict of interest.

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
