# Peer review of "Managerial Practices and (Post) Pandemic Consumption of Private Labels: Online and Offline Retail Perspective in a Portuguese Context"

_sustainability, doi:10.3390/su141710813_

Round 1

Reviewer 1 Report

The paper states to assess the antecedents of the purchase intention of an international food retailer's private label, as well as to explore the impact of the Covid-19 pandemic and the degree of satisfaction and loyalty to this retailer's own brand. To this aim, the authors propose a conceptual model and the related HPs. Empirically, a survey was performed processing data using multiple linear regressions, on a random sample of customers (n=300)

Despite the possible interest for an evergreen topic like consumer behaviour in relation to PLs, the paper presents a number of issues from the theoretical and methodological point of view, requiring a complete rethinking of some HPs and the application of a different method to process data.

From the theoretical viewpoint, is not clear in which way this work contributes in furthering the academic understanding on the topic. I do not see any valuable contribution at the current stage. Moreover, some HPs are questionable. For instance, store satisfaction has been proved to be an antecedent of store loyalty since many years ago (and store image of store satisfaction, see a key paper like Bloemer and Re Ruyter 1998)* and behavioural intention (a proxy of conative loyalty) is expected to play the same role, why in the proposed model it is the inverse. In addition, the literature widely recognises that perceived risk affects attitude. Moreover, the Covid 19 construct is undetermined and I do not believe that the 2 items used to measure it are tested. In sum, the model, the related HPs and the constructs require a strong support and revisions.

*Bloemer, J., & De Ruyter, K. (1998). On the relationship between store image, store satisfaction and store loyalty. European Journal of marketing. Vol. 32 No. 5/6, pp. 499-513

Authors affirm in the abstract that the study “offers a robust and innovative conceptual model for predicting private brand loyalty”. However, the literature review is very basic, the Hps are poorly supported and the method applied to process data is not robust.

In the abstract (row 7) authors state they aim at “understanding the impact of the Covid-19 pandemic on the sustainable purchasing habits of consumers”. This is not a paper dealing with sustainable consumer behaviour so I suggest to refocus the paper’s objective.

In lines 12-14, authors state a mediation effect that is not supported by the analysis performed as a series of regressions can’t verify a mediation role. “Additionally, they reveal that the purchase experience, the image of the private label, the perceived risk and the Covid-19 pandemic are antecedents of attitudes towards the private label, and through their mediation in the purchase intention and recommendation of its products.”

I suggest to the authors to employ structural equation modelling (SEM) to verify their hps in order to get an integrated model and test for mediation. A series of multiple linear regressions are not enough to verify your model and justify publication.

Control variables like demographics and purchase frequency should be taken into accout.

In future papers, please remember to include the HPs numbering and effect (+/-) in the model fig (Fig 1 in this paper)

In future papers, please include the sources of the measurement scales  in the table (table 1 in this paper) reporting the constructs' items (duly validated by previous literature)

The paper requires a professional proofreading.

Author Response

Reviewers’ Comments and Authors’ Response

Sustainability (mdpi) - Special Issue “Consumer Behavior and Sustainable Marketing Development in Online and Offline Settings” [Sustainability]

Title: Managerial practices and (post) pandemic consumption: online and offline retail perspective in the Portuguese context

We sincerely thank both the reviewer and the editor for their compliments and for taking their time to provide such constructive comments and suggestions. We took them into consideration and modified the manuscript accordingly. We believe that the paper has improved significantly.

The updated manuscript includes changes which are highlighted in yellow.

Below you can find our response to each of the comments and suggestions made by the reviewer/editor.

  • We improved our abstract and introduction, in specific about comment “Strongly support your arguments with extant literature. This is particularly needed in the introduction, where you have a long section of text without references, and in fact the whole introduction has only a few of them”. Also, we changed first paragraph of the abstract.
  • “The methodology should be briefly described in the abstract” OK
  • Each of the research hypotheses were supported by extant literature.
  • namely using the authors of the models presented (the combination of the models allowed us to observe the proposed model in its entirety, as a complement of contributions)
  • The manuscript's paragraphs were revised and the ideas were more clearly separated (avoiding, as suggested, very long paragraphs).
  • We added new references:

Sincerely,

The authors

Reviewer 2 Report

-

Author Response

(The authors gave the same response as above.)

Reviewer 3 Report

It is a very good study. Please see my comments, especially on point 7.

11. Please revise the Abstract. It is too long. Make it succinct.

22.  On section 2.1., please change the position of the H1a and H1b, to make them similar to H2a, etc.

33. Line 146 is irrelevant to H4. You may want to modify it.

44. H5a is questionable. The justification is weak.

55.  In section 2.7, there are 2 hypotheses: H7a and H7b. I think it should be: H6a and H6b.

66. Line 203, it should be H7 (not H6). Please see point 5 above. Need more justification this hypothesis.

77.Table 2: The Adjustable R2 of Private Brand Image and Behavioral Intent to the Private Label is too high (above 0.7). It could some issues in developing these hypotheses. Please check again and  modify. 

Author Response

(The authors gave the same response as above.)

Round 2

Reviewer 1 Report

The authors did not reply to any of the concerns raised. Every issue raised and suggestion given were completely ignored. This work presents many inconsistencies and vulnus and is not ready to be published.

Author Response

Title: Managerial practices and (post) pandemic consumption: online and offline retail perspective in the Portuguese context

We sincerely thank both the reviewer (1, 2 and 3) and the editor for their compliments and for taking their time to provide such constructive comments and suggestions. We took them into consideration and modified the manuscript accordingly. We believe that the paper has improved significantly.

The updated manuscript includes changes which are highlighted in yellow.

Below you can find our response to each of the comments and suggestions made by the reviewer/editor.

  • We improved our abstract and introduction, in specific about comment “Strongly support your arguments with extant literature. This is particularly needed in the introduction, where you have a long section of text without references, and in fact the whole introduction has only a few of them”. Also, we changed first paragraph of the abstract.
  • “The methodology should be briefly described in the abstract” -
  • Each of the research hypotheses were supported by extant literature.
  • From the theoretical viewpoint, is not clear in which way this work contributes in furthering the academic understanding on the topic. I do not see any valuable contribution at the current stage. Moreover, some HPs are questionable. For instance, store satisfaction has been proved to be an antecedent of store loyalty since many years ago (and store image of store satisfaction, see a key paper like Bloemer and Re Ruyter 1998)*

*Bloemer, J., & De Ruyter, K. (1998). On the relationship between store image, store satisfaction and store loyalty. European Journal of marketing. Vol. 32 No. 5/6, pp. 499-513. Authors affirm in the abstract that the study “offers a robust and innovative conceptual model for predicting private brand loyalty”. However, the literature review is very basic, the Hps are poorly supported and the method applied to process data is not robust.

  • In the abstract (row 7) authors state they aim at “understanding the impact of the Covid-19 pandemic on the sustainable purchasing habits of consumers”. This is not a paper dealing with sustainable consumer behaviour so I suggest to refocus the paper’s objective. [this comment is very pertinent and we agree. The focus of the manuscript has been improved and revised. Thanks for the suggestions.]
  • namely using the authors of the models presented (the combination of the models allowed us to observe the proposed model in its entirety, as a complement of contributions)
  • In lines 12-14, authors state a mediation effect that is not supported by the analysis performed as a series of regressions can’t verify a mediation role. “Additionally, they reveal that the purchase experience, the image of the private label, the perceived risk and the Covid-19 pandemic are antecedents of attitudes towards the private label, and through their mediation in the purchase intention and recommendation of its products.” We suggest to the authors to employ structural equation modelling (SEM) to verify their hps in order to get an integrated model and test for mediation. A series of multiple linear regressions are not enough to verify your model and justify publication. [We appreciate this comment. We will consider this observation in future research].
  • The manuscript's paragraphs were revised and the ideas were more clearly separated (avoiding, as suggested, very long paragraphs).
  • In future papers, please remember to include the HPs numbering and effect (+/-) in the model fig (Fig 1 in this paper) - we appreciate this comment. We will consider this observation in future research.
  • In future papers, please include the sources of the measurement scales  in the table (table 1 in this paper) reporting the constructs' items (duly validated by previous literature) - we appreciate this comment. We will consider this observation in future research.
  • The paper requires a professional proofreading. The English was proofread by a professional and the manuscript is of higher quality in this version. Thanks for the sugestion.

  • We added new references:
    • Hilken, T., Chylinski, M., Keeling, D. I., Heller, J., de Ruyter, K., & Mahr, D. (2022). How to strategically choose or combine augmented and virtual reality for improved online experiential retailing. Psychology & Marketing39(3), 495-507.
    • Miao, L., Lehto, X., & Wei, W. (2014). The hedonic value of hospitality consumption: Evidence from spring break experiences. Journal of Hospitality Marketing & Management23(2), 99-121.

Sincerely,

The authors

Reviewer 3 Report

You have responded to reviewers' feedback and comments. It is easier to read and follow your paper. Good work.

Author Response

(The authors gave the same response as above.)
